# Teaching Transformers Causal Reasoning through Axiomatic Training

## Abstract

For text-based AI systems to interact in the real world, causal reasoning is an essential skill. Since active interventions are costly to execute, we study to what extent an agent can learn causal reasoning from symbolic demonstrations of causal axioms. Specifically, we consider an axiomatic training setup where an agent learns from multiple demonstrations of a causal axiom (or rule), rather than incorporating the axiom as an inductive bias or inferring it from data values. A key question is whether the agent would learn to generalize from the axiom demonstrations to new scenarios. For example, if a transformer model is trained on demonstrations of the causal transitivity axiom over small graphs, would it generalize to applying the transitivity axiom over large graphs? Our results, based on a novel axiomatic training scheme, indicate that such generalization is possible. For the transitivity axiom, we find that a 67 million parameter transformer model, when trained on linear causal chains (along with some noisy variations) can generalize well to new kinds of graphs, including longer causal chains, causal chains with reversed order, and graphs with branching; even when it is not explicitly trained for such settings. We extend axiomatic training to a harder task of inferring causation from correlation statements and find similar generalization. On both tasks, our model performs at par (or even better) than many larger language models such as GPT-4, Gemini Pro, and Phi-3. Overall, our axiomatic training framework provides a new paradigm of learning causal reasoning in language models that can be extended to arbitrary axioms, as long as sufficient demonstrations can be generated.

## 1 Introduction

Causal reasoning can be defined as a set of reasoning procedures consistent with pre-defined axioms or rules that are specific to causality (Galles & Pearl, 1997). For instance, d-separation and rules of do-calculus (Pearl, 2009b) can be considered as axioms and specifications of a collider or a backdoor set can be considered as rules that can be derived from axioms. Typically, causal reasoning is done over data corresponding to variables in a system. Axioms or rules are incorporated as inductive biases in a machine learning (ML) model, through regularization, model architecture, or the choice of variables for a particular analysis. Depending on the kind of available data—observational, interventional, or counterfactual—Pearl's ladder of causation (Bareinboim et al., 2022) defines the kinds of causal reasoning that is possible.

As axioms are the building blocks of causality, we study whether it is possible to directly learn the axioms using ML models. That is, rather than learning from data that is the result of axioms obeyed by a data-generating process, what if a model can learn an axiom (and thus causal reasoning) directly from symbolic demonstrations of the axiom? This question gains relevance as language models make it possible to learn over symbolic data expressed in natural language. In fact, recent studies have evaluated whether large language models (LLMs) can do causal reasoning by creating benchmarks that encode causal reasoning problems in natural language (Kıcıman et al., 2023; Jin et al., 2024a;b). If we can teach causal axioms to a language model, such a model can be used as a verifier to evaluate output of existing LLMs, or as a reward model to finetune a given LLM for causal reasoning.

Specifically, we propose a new way of learning causal reasoning through axiomatic training. We posit that causal axioms can be expressed as the following symbolic tuple, ⟨*premise*, *hypothesis*, *result*⟩ where *hypothesis* refers to a causal claim and *premise* refers to any relevant information

to decide whether the claim is true or not (*conclusion*). The conclusion could simply be "Yes" or "No". For example, consider the task of inferring causal relationships from correlational statements about a set of variables, which we empirically study in this paper. As in the Corr2Cause dataset from Jin et al. (2024b), the *premise* can be statements about statistical (in)dependence: *"Suppose there is a closed system of three variables: A, B and C. B is correlated with C. A is correlated with C. However, A is independent of B."*; the hypothesis can be a question about cause-and-effect, *"Does A directly cause C?"*; and the *conclusion* would be *"Yes"*. This tuple is a demonstration of the *collider* property/axiom (Pearl, 2009b), which states that if there exist variables such that $A \perp\!\!\!\perp B, B \not\perp\!\!\!\perp C, A \not\perp\!\!\!\perp C$, then there is a unique causal graph connecting them, $A \rightarrow C; B \rightarrow C$. Based on this template, our key insight is that a large number of synthetic tuples can be generated, e.g., by changing the variable names, changing the number of variables, changing the order, and so on. The question is: if a model is trained on such data, would it learn to apply the axiom to new scenarios?

To answer this question, we first train a transformer model from scratch on symbolic demonstrations of the causal transitivity axiom (Galles & Pearl, 1997; Sadeghi & Soo, 2024). To evaluate generalizability, we train on simple chains of the causal irrelevance axiom of size 3-6 nodes and test on multiple different aspects of generalization, including length generalization (chains of size 7-15), name generalization (longer variable names), order generalization (chains with reversed edges or shuffled nodes), and structure generalization (graphs with branching). We find that a model trained on simple chains generalizes to applying the axiom multiple times over larger chains, but it is unable to generalize to the more complex scenarios like order or structure generalization. However, when we train a model on a combined dataset of simple chains and chains with some edges randomly reversed, we find that the model generalizes well across all kinds of evaluation scenarios, including graphs with branching. Our 67 million parameter model outperforms billion-scale models like Gemini Pro, Phi-3, and in some cases GPT-4, for both zero-shot and multi-shot settings. Extending the findings on length generalization for NLP tasks (Kazemnejad et al., 2023; Bhattamishra et al., 2020; Haviv et al., 2022; Furrer et al., 2021), we find a critical role of positional embedding in ensuring causal generalization across length and other aspects. Our best model has no positional encoding, although we find that sinusoidal encoding also works well for some scenarions.

Baed on these results, we apply the axiomatic training approach to the problem of inferring causality from correlational statements under the Corr2Cause setting (Jin et al., 2024b), as described above. Using the same method to generate synthetic training data and train the model, we find that a transformer trained on task demonstrations over 3-4 variables learns to solve this task for graphs with 5 variables. On this task too, our model's accuracy is higher than larger LLMs such as GPT-4 and Gemini Pro, even when multi-shot examples are provided to the LLMs.

Our work provides a new paradigm of teaching models causal reasoning through symbolic demonstrations of axioms, which we call *axiomatic training*. Such symbolic data can be cheaply generated for multiple axioms and added to the pretraining or finetuning data for language models, as done by Papadimitriou & Jurafsky (2023) for language structure constraints. The data generation and training procedure is general and can be applied to learn any new axiom (e.g., logical axioms), as long as it can be expressed in the symbolic tuple format. More generally, our results contribute to the literature on causal learning from *passive data* (Lampinen et al., 2023), showing a general way to learn any causal axiom through passive demonstrations.

## 2 RELATED WORK

**LLMs for Knowledge-Driven Causal Reasoning:** Recent developments in Large Language Models (LLMs) have highlighted their potential for knowledge-driven causal discovery. Unlike traditional methods which focus on statistical patterns or correlations, LLMs utilize knowledge acquired through their pretraining to reason about and identify causal structures based on metadata of variables (Kıcıman et al., 2023; Ban et al., 2023; Long et al., 2023; Willig et al., 2022; Vashishtha et al., 2023). However, possibility of memorization of existing benchmarks in the pretraining of these LLMs has been a major criticism. As a result, recent work (Zečević et al., 2023) argues that LLMs are not actually performing causal reasoning, but simply learning correlations about causal facts. In addition, there are critical failure modes of using LLMs for causal discovery due to hallucinations or not obeying the acyclic constraint when generating graph edges (Vashishtha et al., 2023). To evaluate

causal reasoning capabilities of LLMs, (Jin et al., 2024b) and (Jin et al., 2024a) propose formal causal inference evaluation benchmarks to infer direct and indirect causal relationships, and highlight the failure of LLMs in performing accurate causal reasoning.

**Impact of Positional Encoding on Generalization:** Length generalization capabilities of transformers has been studied in the past to better understand their different failure modes across various settings (Hupkes et al., 2020; Zhang et al., 2023; Furrer et al., 2021). Previous work (Kazemnejad et al., 2023; Bhattamishra et al., 2020; Haviv et al., 2022; Shen et al., 2023) emphasizes the impact of positional encoding in length generalization capability of transformers. To understand how transformers can be optimized for learning through axiomatic training and generalizing to unseen larger causal structures, we also examine different types of positional encoding such as no positional encoding (PE), Learnable PEs (Radford et al., 2018) and Sinusoidal PEs (Vaswani et al., 2023).

**Synthetic data generation for teaching transformers reasoning:** Synthetic data generation has been explored for optimising model training for reasoning. For example, (Li et al., 2023; Gunasekar et al., 2023) use LLM-generated synthetic text for training Phi-1 and Phi-1.5 models and show impressive performance for reasoning-based tasks. (Trinh et al., 2024) introduce a novel neuro-symbolic framework to pre-train a transformer model for Olympiad-level math problems. Building on this stream of work, we apply synthetic data generation for teaching causal reasoning.

## 3 LEARNING CAUSAL AXIOMS USING TRANSFORMERS

Instead of performing causal reasoning using observational or interventional data, we study whether it is possible to learn general rules of causality directly from symbolic axioms. We begin by asking the question "are there any minimal sufficient characterization of causal principles?". There has been fundamental work from Galles & Pearl (1997) where they axiomatize causal relevance (or equivalently irrelevance). They show that for a given *stable probabilistic* causal model (defined below), there exists a finite set of axioms that are completely characterized by axioms of path interception in corresponding directed graphs. We study how such causal relevance statements can be incorporated into transformer models. Throughout this work, we assume no unobserved confounders.

**Notation.** We denote a random variable with the upper case letter letters (e.g. $X, Y, Z$ and use lower case letters (e.g. $x, y, z$) to denote the value taken by the corresponding random variable denoted as $X = x, Y = y, Z = z$. We represent the probability of a random variable using $X_i$ by $\mathbb{P}(X_i)$. Let $\mathcal{G}(\mathbf{X}, \mathbf{E})$ be a directed acyclic graph (DAG) consisting of a set of variables $\mathbf{X} = \{X_1, \ldots, X_n\}$ and a set of directed edges $\mathbf{E}$ among variables in $\mathbf{X}$. Let $pa(X_i) = \{X_k | X_k \to X_i\}$, $de(X_i) = \{X_k | X_k \leftarrow \cdots \leftarrow X_i\}$, $ch(X_i) = \{X_k | X_i \to X_k\}$ denote the set of *parents*, *descendants* and *children* of $X_i$ respectively. Next, we define some special notations in the context of three variables. Given two nodes $X_i, X_j$ we call a third node $X_k$ to be a *collider* if both $X_i$ and $X_j$ are parents of $X_k$. $X_k$ is called a *mediator* if one of the nodes, say, $X_i$ is parent and the other node $X_j$ is child of $X_k$. Lastly, the $X_k$ is called a common cause of both $X_i$ and $X_j$ are children of $X_k$.

**Definition 3.1** (**Causal Irrelevance**, adapted from Defn. 7 in (Galles & Pearl, 1997)). *X is probabilistically causally irrelevant to Y given Z, written $(X \nrightarrow Y | Z)$ iff: $\mathbb{P}(y | z, do(X) = x) = \mathbb{P}(y | z, do(X) = x')$, $\forall x, x', y, z$ i.e., once we hold Z fixed at z, intervening on X will not change the probability of Y.*

Next, we restate the stability assumption for a causal model from Galles & Pearl (1997) that gives a richer set of finite axiomatization for probabilistic causal irrelevance.

**Assumption 3.1** (Stability, Definition 9 in Galles & Pearl (1997)). *Let $\mathcal{M}$ be a causal model. Then an irrelevance $(X \nrightarrow Y | Z)$ in $\mathcal{M}$ is stable if it is shared by all possible probability distribution over $\mathcal{M}$. The causal model $\mathcal{M}$ is stable if all of the irrelevances in $\mathcal{M}$ are stable.*

Under the stability assumption (see Assumption 3.1), Galles & Pearl (1997) states six axioms that completely characterize causal irrelevance (Definition 3.1) via axioms of path interception in the directed graphs. An axiom of causal irrelevance is of the form (given in conjunctive normal form):

$$\bigwedge_s \bigvee_t (X_i^{s,t} \nrightarrow X_j^{s,t} | X_k^{s,t}) \implies \bigwedge_l \bigvee_n (X_i^{l,n} \nrightarrow X_j^{l,n} | X_k^{l,n})$$

where $\wedge$ is "logical and", $\vee$ is "logical or" and for a given $(s, t)$ or $(l, n)$ pair, $\boldsymbol{X}_i, \boldsymbol{X}_j, \boldsymbol{X}_k$ are disjoint subsets of observed variables $\boldsymbol{X}$. In the above causal irrelevance statement, if the antecedent is true, the consequent is also true.

**Transitivity Axiom.** We illustrate our axiomatic training procedure through the transitivity axiom. Following the stability assumption above, we consider the class of interventional distributions in which the transitivity causal irrelevance axiom holds (Sadeghi & Soo, 2024). Formally, for a stable probabilistic causal model (§3), given variables $X, Y, Z$ in the system, the transitivity axiom is:

$$(X \nrightarrow Y|Z) \Rightarrow (A \nrightarrow Y|Z) \vee (X \nrightarrow A|Z) \forall A \notin X \cup Z \cup Y \tag{1}$$

which can be simplified using the contrapositive.

$$\exists A \notin X \cup Y \cup Z \ \ s.t. \underbrace{(X \to A|Z) \wedge (A \to Y|Z)}_{P:\text{premise}} \implies \underbrace{(X \to Y|Z)}_{H:hypothesis}$$

We call the LHS as *Premise* and the RHS as *Hypothesis*. Our key idea is that we can use such an axiom to generate thousands of synthetic symbolic expressions that can be used to teach a transformer the specific axiom. The trained model is then evaluated on whether it can apply these axioms to new causal structures that were not available in the training set. In all our experiments, we consider an empty conditioning set $Z$ for simplicity.

### 3.1 AXIOMATIC TRAINING: DATASET, LOSS FUNCTION, AND POSITIONAL ENCODING

**Training data.** Based on a specific axiom, we can map a hypothesis given the premise to its correct label ('*Yes*' or '*No*'). To create a training dataset, we enumerate all possible tuples of $\{(P, H, L)\}_N$ where $P$ is the premise, $H$ is the hypothesis and $L$ is the label *(Yes/No)* for a particular setting of the variables $X, Y, Z, A$. Given a premise $P$ based on a given causal graph, if the hypothesis can be derived by applying the specified axiom (once or multiple times), then label $L$ is *Yes*; otherwise, *No*. For example, for the transitivity axiom, suppose the underlying true causal graph of a system is a chain, $X_1 \to X_2 \to X_3 \to \cdots \to X_n$. Then, a possible premise could be $X_1 \to X_2 \wedge X_2 \to X_3$, and the corresponding hypothesis $X_1 \to X_3$ will have label *Yes* whereas another hypothesis $X_3 \to X_1$ will have label *No*. The above axiom could be inductively applied multiple times to generate more complex training tuples.

**Loss function.** Given a dataset, the loss function is defined based on the ground truth label for each tuple, represented as $\mathbb{E}_{P,H,L \sim P_{\text{train}}} - \log(P(L|P, H))$. A preliminary analysis indicated promising results with this loss formulation compared to next token prediction loss.

**Positional Encoding.** In addition to the training data and loss function, recent work (Kazemnejad et al., 2023) has shown that the choice of positional encoding is important for generalizing a transformer to longer or complex inputs. Therefore, we experiment with different positional encoding to understand their impact on generalization in causal tasks: learnable position encoding (LPE), sinusoidal positional encoding (SPE), no positional encoding (NoPE). See Appendix D for details.

### 3.2 DATA PERTURBATION: THE KEY TO MODEL GENERALIZATION

To test generalization, we train the model on simple causal structures like sequential chains and evaluate its performance on more complex structures. To enhance generalization, we introduce structured perturbations in the training data across three dimensions: variable names, causal structure types, and the number of variables.

1. **Node names**: Each node in the graph is represented by an alphanumeric name comprising 1-3 characters. The length of a name and the specific characters are randomly selected during data generation.
2. **Causal Graph Topology**: For the transitivity axiom, we consider two main types of causal graphs in the training set.
   (a) **Sequential**: All causal edges are directed forward, thus forming a typical transitivity chain, e.g. $X \to Y \to Z$.
   (b) **Random Flipping**: Given a chain of sequential nodes, we randomly reverse some edges creating complexity by disrupting direct paths between subsequent nodes (eg. $X \to Y \leftarrow Z$). This can be expressed simply through natural language like: *"X causes Y. Z causes Y."*

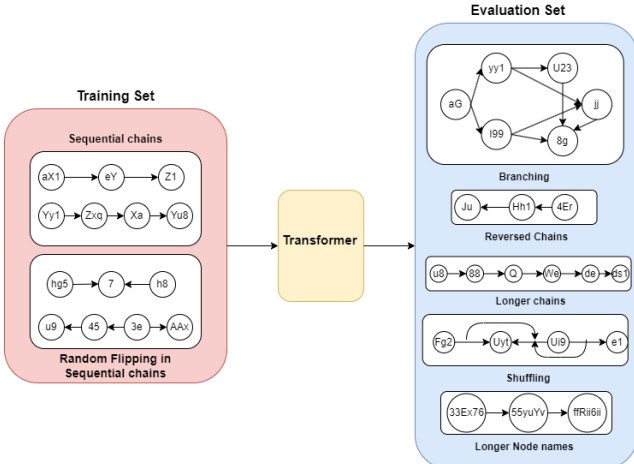

Figure 1: **Evaluating structural generalization of transformers through axiomatic training.** We train a transformer on two simple causal structures: chains and chains with random flipping of some edges. All training instances consist of 3-6 nodes. The trained model is evaluated on significantly more complex structures: bigger causal chains with >6 nodes, general branched networks with higher average in-degree and out-degree, complete reversals, longer sequences, shuffled natural language statements of sequences and longer node names.

3. **Length level**: To facilitate transformers understanding of the axiom, we incorporate chains of varying lengths, ranging from 3 to 6 nodes in our training set.

Random flipping introduces forks and colliders, which form the building blocks of any causal DAG. This helps add complexity in model training, thus aiding generalization across multiple structures.

### 3.3 Evaluation setup: Assessing Axiomatic Learning in Transformers

To evaluate if a trained model has learnt the correct understanding of an axiom instead of shortcuts or correlation-based features, designing an out-of-distribution (OOD) evaluation set is important. We evaluate our model on multiple types of complex structures that are unseen during training.

1. **Length**: Evaluating whether our model accurately infers causal relationships for sequences or chains (both sequential and ones with random flipping) longer than those in the training set.
2. **Node Name Shift**: Testing the model's performance on longer node names, from 1-3 characters used in the training set to 8-10 characters. This is motivated by Jin et al. (2024b) who show how change in node names results in generalization failure on causal tasks for language models.
3. **Order of Chains**: a) **Completely reversed chains**: This evaluation is inspired by the reversal curse (Berglund et al., 2024) that revealed generalization failure of LLMs in answering questions in reversed sequences despite knowing the answers in the original order. We evaluate our model on completely reversed chains, a structure that was not in the training data. A completely reversed chain will be of the form $X \leftarrow Y \leftarrow Z$, written in natural language as: *"Y causes X. Z causes Y."*, where $X, Y, Z$ are replaced by random alphanumeric names. b) **Shuffling of Sequences:** Causal sequences with random edge flips, as defined in 3.2 represented by natural language statements sequentially *(A causes B. B causes C ...)*, are shuffled to add complexity and break sequential order. This tests model's ability to infer accurate relationships regardless of sequence order.
4. **Branching**: We also evaluate on complex graphs beyond chains, measured using the branching factor: Number of edges/Number of nodes. While the training set comprises simplistic sequences, this evaluation setup involves multiple branches, colliders, forks, and chains in one network, thus having significantly high complexity.

## 4 Application 1: Axiomatic Training For Transitivity Axiom

### 4.1 Training and Evaluation Datasets

For learning the transitivity axiom, a synthetic dataset $D$ is constructed with $N$ axiomatic instances generated using the transitivity axiom. Each instance in $D$ is structured in the form of a premise $P$, which is the natural language expression of a causal structure (e.g., *"X causes Y. Y causes*

$Z$"), followed by the hypothesis in the form of a question $H_q$ (e.g., "Does $X$ cause $Y$?"), which is then followed by the final label $L$ (e.g., "Yes" or "No"). Formally, each premise $X_i$ takes the form: $X_i = [e_{jk} \mid j > 0, k > 0]$. Here $e_{jk}$ represents an edge between node $j$ and $k$ in the $ith$ causal sequence (can also be represented as $X_{i,j}$, $X_{i,k}$) such that each $e_{jk}$ translates to "$X_{i,j}$ causes $X_{i,k}$." in natural language.) For each premise, all possible hypotheses consisting of two variables are generated (i.e., "Does A cause B" and "Does B cause A" for each pair). The training data consists of graphs of lengths from [3,6], with branching factor [0.6, 0.8], and length of nodes [1,3]. In addition to sequential chains, random flipping of edges is done with 0.5 probability. See Appendix E for details on these hyperparameters.

Our training data consists of 175k axiom demonstrations. We use three versions of training data to evaluate the impact of different data perturbations, each with equal number of 'Yes' and 'No' labels.

- **Training Setup 1 (TS1)**: This setup comprises of 73k chains with random flipping and 101k sequential linear chains. Since flipping is done randomly across all consecutive pairs of nodes in the given chain, some complete reversals are also formed. In this training set around 12k completely reversed chains are present. However, only when evaluating for reversal chain test setup, we remove all the reversed chains from this training set and re-train the model from scratch with the remaining sequential and randomly flipped chain sequences.
- **Training Setup 2 (TS2)**: This setup comprises of more simple sequential chains (132k), while we decrease chains with random flipping (42k), to keep the overall size around 175k.
- **Only Causal Chains (OCC)**: This set comprises of sequential transitivity chains without any edge being randomly flipped, to help understand whether adding perturbations helps for generalization.

**Evaluation Datasets.** We use the following evaluation datasets to assess model generalization.

- **Length Generalization EvalSet:** Testing on causal sequences with length >6 upto 15, longer than any sequence encountered by the model in training set. Length generalization is evaluated for both sequential chains and chains with randomly flipped edges.
- **Node Name EvalSet:** Assessed model's generalization capabilities to sequences with longer node names, increasing from 1-3 characters used in training set to 8-10 characters. To add onto the complexity, we also include sequences longer than any sequence in the train set (>6) upto 9 length.
- **Reversal EvalSet:** Evaluated performance of our transformer model, with no completely reversed sequence in its training, on reversed causal sequences. Sequences upto 6 length were evaluated.
- **MultiEval$_{\text{SLR}}$ (Shuffling + Random Flipping + Length Sequence)**: This setup involves evaluation on 3 levels of complexities together: shuffling of sentence for representing the sequences, each sequence having random flipping, and some sequences having longer length than sequences in training set (upto 9).
- **Branching EvalSet:** One of the most complex evaluation setups, with dense networks containing multiple branches, colliders and forks. While each sequence in the training set had values of 1-2 for both in-degree and out-degree across all nodes, in this setting a node can have maximum value of $n-1$ for both, and minimum of 0 creating more complicated structures than the ones transformer had encountered during its training. To add onto the complexity we evaluate on structures with more nodes (8,10,12), than any unique causal sequence in the training set besides 5 node networks. We evaluate multiple densely branched networks constructed using the Erdös-Rényi model, where we provide number of edges and nodes in accordance to the values of branching factor (1.4 and 2) we use for evaluation. We implement this using igraph package in python Csardi & Nepusz (2006) to get different unique graphs with required branching factors for evaluation.

### 4.2 IMPLEMENTATION DETAILS: ARCHITECTURE, TOKENIZER AND TRAINING PROCEDURE

We train a decoder-based 67 million parameter model based on GPT-2's architecture. The model has 12 attention layers, 8 attention heads and 512 embedding dimensions. The model is trained from scratch on each of our training datasets. To understand the effect of Positional Encodings (PE), we consider Sinusoidal PE (SPE) (Vaswani et al., 2023), Learnable PE (LPE) (Radford et al., 2018) and having no PEs (NoPE) (Kazemnejad et al., 2023; Haviv et al., 2022). All models are trained for 100 epochs using the AdamW optimizer with 1e-4 learning rate.

Since the training dataset follows a specific structure, we develop a custom tokenizer. Alphanumeric node names are tokenized at a character level, while special terms such as *'causes', 'Does', 'cause',*

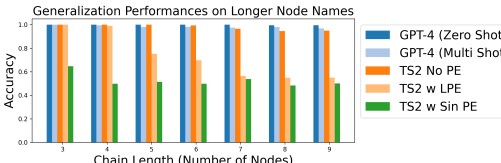 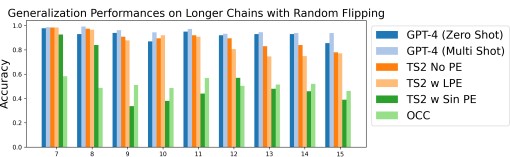

Figure 2: Evaluating generalization on causal sequences (without random flipping) with longer node names (than the ones used in sequences in train set). TS-2 training set with no positional encoding leads to the best performance. Refer table A4 for complete results.

Figure 3: Generalizing to longer unseen causal sequences (>6 nodes) with random flipping on TS2 and OCC (with NoPE) train sets. OCC-trained models struggle due to limited edge-level variability, while TS2 NoPE consistently performs well. Refer table A3 for complete results

*'Yes'*, and *'No'* are tokenized at the word level. Such an approach avoids out of vocabulary (OOV) tokens at test time since the alphanumeric node names in the test set can be different than those in the training set. Following this approach, the vocabulary size of our transformer model is 69.

## 4.3 BASELINES USING EXISTING LLMs

Given recent work on how LLMs can be leveraged for causal reasoning (Kıcıman et al., 2023; Vashishtha et al., 2023; Ban et al., 2023), we include language models such as GPT-4 *(gpt-4-32k)* ope (2024), Gemini *(gemini-pro)* gem (2024) and Phi-3 *(Phi-3-mini-128k-instruct)* abd (2024) as baselines. Note that each of these models is significantly larger than our model and known to perform well on reasoning tasks, with the smallest baseline model Phi-3 having 3.8 billion parameters.

**Zero Shot Setting** To evaluate the baseline models, we follow a simple zero-shot prompting strategy. For each tuple, we provide the natural language expression of the causal graph *(Premise)* followed by the question *(Hypothesis)* and prompt the LM to answer it in either 'Yes' or 'No' *(Label)*. Here is an example prompt: *"EX causes T. T causes 9. 9 causes W. W causes 7. 7 causes M. M causes a. Does EX cause T? Answer in 'Yes' or 'No' only."* Table A1 contains some examples of all types of prompts and instances used for querying as well as training for different structural types.

**Multi Shot Setting** We present few-shot instances from our training data that include sequential causal chains, along with a few examples with random flipping of edges. All multi-shot instances were sourced exclusively from the training set. Each example in the prompt follows the same *Premise-Hypothesis-Label* structure as explained above. A.1 contains the multishot prompt created from train set and used for querying baseline LLMs.

## 4.4 RESULTS: GENERALIZATION TO COMPLEX CAUSAL SCENARIOS

We present results on how well an axiomatically trained transformer can generalize to larger and more complex causal graphs, and how it compares to pre-trained LLMs.

**Length Generalization:** Table A3 shows accuracy of different models when evaluated on larger causal chains that were not seen during training. Among the baseline pre-trained LMs, GPT-4 obtains the highest accuracy on both standard and randomly flipped chains for the multi-shot setting. It is remarkable that our TS2 (NoPE) model obtains competitive performance to the trillion-scale GPT-4 model, even though it had never seen larger sequences (length 6) during training. In particular, for chains of size 7-12, TS2 (NoPE) obtains higher or comparable accuracy than GPT-4 across the standard and randomly flipped chains. Similar trends are observed for chains of size 7-13 when compared to GPT-4 in the zero-shot setting. Our model's accuracy decreases for chains of length 14-15 (0.85 for standard chains and 0.78 for randomly flipped chains) but is still significantly higher than that of LMs like Gemini-Pro and Phi-3. Although in-context examples improve the performance of baseline LLMs, TS2 (NoPE) still easily outperforms both Gemini Pro and Phi-3 in the multi-shot setting. Note that a random prediction would yield a 50% accuracy, indicating that the axiomatically-trained TS2 (NoPE) model can generalize its reasoning to causal chains much longer than 6 even though it was trained only on chains upto length 6.

**Node Name Shift:** For models trained on `TS2` dataset, we also evaluate generalization to changes in variable names (Figure 2). We find that `TS2` (`NoPE`) is robust to node name changes and retains its high accuracy as new, longer names are introduced. It also retains its generalizability to longer sequences with new node names, performing similarly to GPT-4.

**Order of Causal Sequences:** We now consider how variations in the causal structure impact generalization of axiomatically-trained models. In Table 1b, we consider the complex evaluation setup **MultiEval$_{\mathbf{SLR}}$** that includes shuffled order of causal sequences with random flipping for increasing length (even beyond the ones in train set).While GPT-4 performs best, models trained with LPE and NoPE still achieve strong results, surpassing Gemini Pro and Phi-3 in both zero-shot and multi-shot settings. On this task, `TS2` (`NoPE`) achieves higher accuracy than Gemini Pro and Phi-3 on chains up to length 8. At length 9, `TS2` (`NoPE`) reaches 0.73 accuracy, comparable to Gemini Pro (0.74) and much better than random. A similar trend is seen for completely reversed sequences (Table 1a). This task presents extreme out-of-distribution data, as the training data contains left-to-right edges, while the test data has only right-to-left edges. Our axiomatically trained model `TS2` (`NoPE`) outperforms GPT-4 (zero-shot) on chains of length 3-6. Even though baseline LLMs improve in multi-shot settings, `TS2` (`NoPE`) consistently outperforms Gemini-Pro and Phi-3, and remains competitive with GPT-4. In particular, its accuracy (0.94 for chains of length 6) is substantially higher than Gemini Pro and Phi-3 (0.62 and 0.69 respectively).

**Branching:** Finally, we consider the structurally hardest evaluation task involving non-linear chains where we introduce general Erdos-Renyi graphs as the causal sequences while the training data contains only linear chains A2. Here the size of network corresponds to the number of nodes and branching factor defined in 4.1 in the graph and we study the performance differences as the branching factor is varied. While GPT-4 achieves the highest accuracy as graph sizes increase, our `TS2` (`NoPE`) model outperforms Gemini Pro (branching factor 1.4) in all but one graph size under zero-shot settings. On graphs with 12 nodes and a 1.4 branching factor, `TS2` (`NoPE`) achieves 70% accuracy, far better than random (50%), despite training only on graphs with branching factors 1. Although LLMs excel in multi-shot settings, our model's performance shows how simple structural perturbations during training can enhance out-of-distribution generalization and reasoning capabilities.

**Summary:** Across all evaluation setups, our axiomatically trained model `TS2` (`NoPE`) performs significantly better than random baselines even as chain lengths are increased beyond its training data. In particular, even though our model was not trained on fully reversed chains, it performs at par with the significantly larger GPT-4 model (Fig. A2), while easily outperforming other billion scale models even under multi-shot settings. For other tasks, it often outperforms or matches the accuracy of billion-scale models like Gemini Pro and Phi-3. These results indicate that a model trained axiomatically can learn to reason about more complex causal structures from demonstrations of simple causal sequences.

### 4.5 Additional Results: Role of Data Diversity and Positional Encoding

**Importance of Data Perturbations.** We find that diversity of the sequences in train data plays an important role. Model trained on only causal chains (OCC) generalize to longer chains (Table A3) but not to other DAG structures (see Figure 3 for edge flip, Figure A2 for reversal, Table A2 for branching). Models trained on TS1 or TS2 generalize across all scenarios, including random flip, order permutations, and branching; thus highlighting the impact of incorporating variability at the edge level through random flipping. However, across tasks, we find that `TS2` yields higher accuracy than `TS1`, even as `TS1` has more variations due to random flipping. This suggests that while perturbations aid structural generalization, excessive perturbations can hinder it (in particular, random flipping may decrease the length of available causal paths during training).

**Role of Positional Encodings.** When comparing models based on positional encoding, we find that models without positional encoding generalize well to both longer chains (up to length 15) and unseen complex graph structures, despite being trained only on linear chains with 3-6 nodes. Models with SPE and LPE perform well on longer chains but struggle with longer node names, even in smaller graphs (Figure 2), highlighting their sensitivity to minor perturbations. SPE also underperforms in branching and order-based settings like shuffling and reversal. Learnable PE works up to 9-length chains but drops afterward. Overall, our results extend earlier work on the utility of `NoPE` (Kazemnejad et al., 2023; Haviv et al., 2022) to the task of understanding causal sequences and generalizing to both longer length and complex structure at test time. Interestingly, all PEs perform

Table 1: Reversal and Shuffling

| Model | 3 | 4 | 5 | 6 |
|---|---|---|---|---|
| **Baselines** | | | | |
| **Zero Shot** | | | | |
| GPT-4 | 0.97 | 0.99 | 0.98 | 0.92 |
| Gemini Pro | 0.61 | 0.59 | 0.66 | 0.62 |
| Phi-3 | 0.80 | 0.69 | 0.73 | 0.69 |
| **Multi Shot** | | | | |
| GPT-4 | **1.00** | **1.00** | **1.00** | **0.99** |
| Gemini Pro | 0.95 | 0.87 | 0.77 | 0.71 |
| Phi-3 | 0.93 | 0.89 | 0.75 | 0.75 |
| **Axiomatic Training** | | | | |
| TS1 w NoPE | 0.98 | 0.99 | 0.92 | 0.91 |
| TS1 w SPE | **1.00** | 0.99 | 0.99 | 0.97 |
| TS2 w NoPE | 0.99 | 0.99 | 0.95 | 0.94 |
| TS2 w SPE | 0.98 | 0.97 | 0.93 | 0.94 |
| TS2 w LPE | 0.99 | 0.98 | 0.95 | 0.97 |
| OCC w NoPE | 0.33 | 0.18 | 0.10 | 0.09 |

(a) Following (Berglund et al., 2024), we evaluate models on inferring cause-and-effect from fully reversed sequences absent in training data. Models trained on OCC perform worse, highlighting the importance of edge-level perturbations for generalization. Accuracy metric is reported, with random baseline = 0.5. Best performance is bolded, while second best is underlined.

| Model Config | 3 | 4 | 5 | 6 | 7 | 8 | 9 |
|---|---|---|---|---|---|---|---|
| **Baselines** | | | | | | | |
| **Zero Shot** | | | | | | | |
| GPT-4 | 0.99 | 0.97 | 0.89 | 0.85 | **0.95** | **0.90** | 0.90 |
| Gemini Pro | 0.75 | 0.73 | 0.72 | 0.76 | 0.71 | 0.68 | 0.74 |
| Phi-3 | 0.88 | 0.86 | 0.82 | 0.79 | 0.76 | 0.73 | 0.79 |
| **Multi Shot** | | | | | | | |
| GPT-4 | **1.00** | **0.99** | **0.97** | **0.95** | 0.94 | **0.90** | **0.92** |
| Gemini Pro | 0.95 | 0.85 | 0.83 | 0.79 | 0.79 | 0.73 | 0.75 |
| Phi-3 | 0.88 | 0.83 | 0.82 | 0.80 | 0.83 | 0.76 | 0.78 |
| **Axiomatic Training** | | | | | | | |
| TS1 NoPE | **1.00** | 0.94 | 0.87 | 0.84 | 0.80 | 0.76 | 0.73 |
| TS1 LPE | **1.00** | 0.95 | 0.87 | 0.83 | 0.78 | 0.78 | 0.71 |
| TS1 SPE | **1.00** | 0.94 | 0.86 | 0.83 | 0.76 | 0.73 | 0.68 |
| TS2 NoPE | **1.00** | 0.95 | 0.87 | 0.84 | 0.79 | 0.76 | 0.73 |
| TS2 w LPE | **1.00** | 0.94 | 0.87 | 0.84 | 0.80 | 0.76 | 0.73 |
| TS2 w SPE | 0.99 | 0.94 | 0.89 | 0.84 | 0.75 | 0.74 | 0.49 |
| OCC w NoPE | 0.69 | 0.62 | 0.57 | 0.54 | 0.57 | 0.53 | 0.52 |

(b) Evaluated on **MultiEval$_{SLR}$** setup defined in 4.1. Models trained on TS1 and TS2 with NoPE and LPE perform well across all sequence lengths (even those which were not in the train set) while models trained with SPE or on OCC face generalization failures. Accuracy metric is reported wherein random baselines would yield 0.5. Best performance is bolded, while second best is underlined.

well in randomly flipped sequences, likely due to the short effective path lengths caused by the 0.5 probability of forward-directed edges.

## 5 APPLICATION 2: INFER CAUSATION FROM CORRELATION STATEMENTS

While the above study evaluated transformers' capability to generalise the transitivity axiom from small causal chains to large graphs, we now study whether this capability transfers to a more general causal task. To this end, we apply axiomatic training to a task on inferring causation from statements about correlation in observational data (Jin et al., 2024b). Each data instance in the benchmark includes correlational relationships described in natural language for graphs with 3 to 6 nodes; the goal is to infer the truth value of a hypothesis (of six types: Parent, Ancestor, Child, Descendant, Collider, Confounder) among a set of variables. This task is significantly harder than applying the transitivity axiom. First, there are multiple hypothesis types to evaluate: direct effect, indirect effect, children, ancestors, colliders and confounders. Second, solving the task requires an understanding of d-separation (refer d-separation in section C) and the **Markov property** (refer C.1). Specifically, it involves mapping correlational statements to multiple possible causal graphs and determining if the query is satisfied across all graphs in the **Markov Equivalence Class** (refer C.1).

**Task example:** To infer causal relationships from correlational statements, we use d-separation to determine conditional independence, which helps infer the causal graph's skeleton. For example, given A → B → C, we infer that A is independent of B given C. With this, a model can evaluate hypotheses, like "Is B a collider?", and conclude that it's false, as the structure is either a chain or a fork. The task follows a Premise-Hypothesis-Label structure, where the premise are the correlational statements, the hypothesis asks relationships between nodes, and the label provides final answer.

**Model Training:** We use the same model architecture as in Section 4.2 and train our model from scratch for 100 epochs using NoPE, since it performed consistently well across diverse OOD settings in our transitivity based experiments. For creating a train set, we consider the subset of the original dataset with correlational statements for graph consisting of 3 and 4 nodes. As the test set, we evaluate the model's performance directly on 5 node correlational statements.

To aid generalization, we take inspiration from our transitivity-based experiments and create different combinations of randomly created alphanumeric node names. We then derive a training set from the original dataset by instantiating the correlational statements with different combinations of alphanumeric node names. We balance the dataset by sampling equally from both classes to avoid bias in our transformer model to get a train set with 113k instances. Then, we create a test set with 1000 randomly sampled instances of correlational statements for 5-node graph networks. Since the correlational statements are not simplistic unlike the premise from our transitivity experiments, we tokenize at the character level for nodes. For a straightforward evaluation, we tokenize all input text at the token level and use the same node names for evaluation as in the training set to avoid potential out-of-vocabulary issues.

| Model | Precision | Recall | F1 Score | Accuracy |
|---|---|---|---|---|
| **Ours** | **0.72** | 0.50 | 0.59 | **0.64** |
| **Zero-Shot** | | | | |
| Phi-3 | 0.52 | 0.60 | 0.56 | 0.52 |
| Gemini pro | 0.52 | 0.59 | 0.55 | 0.52 |
| GPT-4 | 0.59 | 0.50 | 0.54 | 0.58 |
| **Multi-Shot** | | | | |
| Phi-3 | 0.57 | 0.67 | 0.61 | 0.58 |
| Gemini pro | 0.51 | **0.74** | 0.60 | 0.52 |
| GPT-4 | 0.66 | 0.56 | **0.61** | **0.64** |

Table 2: Correlation to Causation Experiments adapted from (Jin et al., 2024b). Axiomatic training setup aids generalization even for complex causal tasks, while bigger LLMs struggle on the same in zero-shot setting. Refer section 5 for details regarding experimental setup and result trends.

**Comparison with Baselines:** As reported in (Jin et al., 2024b), pre-trained LMs perform similarly to random guessing. Phi-3 and Gemini Pro have a similar perfrormance (52% accuracy) in zero-shot settings (see Figure A1 and Table 2). GPT-4 performs slightly better (58%) but shows a significant improvement in multi-shot. Remarkably, our small transformer outperforms all zero-shot baselines with 64% accuracy, 6% higher than GPT-4 in zero shot. Even in multi-shot settings, our model matches GPT-4, suggesting that axiomatically-trained transformers could be further optimized for causal reasoning tasks.

## 6 DISCUSSION AND CONCLUSION

In this work, we provide a method to create training data containing diverse demonstrations of an axiom and explore modelling choices to learn the axiom. A transformer model trained from scratch on a large axiomatic dataset can learn to apply axioms effectively. On causal tasks like graph traversal via transitivity and inferring causal relationships from correlation, small 67M transformers generalize well to unseen complex graphs, often outperforming models like GPT-4, Phi-3, and Gemini Pro. Extending to more complex tasks of (Jin et al., 2024b), our robust training enables the small model to match or exceed the performance of larger models.

**Applicability to Causal Tasks.** While our current work focuses on the transitivity axiom for causal relevance, extending the work to other causal axioms from (Galles & Pearl, 1997) is an interesting research direction. In addition, we may consider other axioms that are relevant for downstream tasks. For example, if a transformer model can be trained to validate the d-separation rule—given two variables X and Y, are they independent given a variable set Z?, then repeated applications of the rule can be used to derive a valid backdoor set.

**Generalization to Logical Reasoning.** While our axiomatic training approach focuses on causal reasoning, it can be applied to any formal system such as deductive logical reasoning. Recent work (Saparov et al., 2023) highlights LLMs' deterioration in performance as reasoning depth increases. It would be interesting to explore if axiomatic training can improve deductive reasoning in LMs, given the similarity between our setup and such tasks.

**Implications for Training Language Models.** GPT-4 demonstrates impressive generalization on the causal tasks we evaluated. We hypothesize that axiomatic training may explain GPT-4's ability to reason over causal graphs, as (noisy) demonstrations of the underlying axioms could be present in its web-scale training data. Meanwhile, models like Gemini Pro and Phi-3 struggle with zero-shot reasoning for causal tasks, such as handling completely reversed chains, suggesting room for improvement. Incorporating causal axiom demonstrations as a part of language models' pretraining (or finetuning data) could help improve the reasoning of these models, so that small language models like Phi-3 can achieve GPT4-like accuracy on causal tasks. Incorporating axiomatic inductive biases could aid language models reasoning abilities for the desired task. For instance, (Papadimitriou & Jurafsky, 2023) propose pretraining language models on synthetic formal languages to incorporate inductive biases, to improve performance on learning grammatically diverse languages.

## ETHICAL IMPACT AND REPRODUCIBILITY

*Ethical Statement.* All datasets used in our work will be made publicly available for evaluation of future iterations of models. We made best efforts to compare against contemporary models in a fair manner. There may be no direct harmful impact, especially considering our work focuses on a synthetic symbolic setting. However, since LLMs may be used in our approach, suitable prudence may be necessary to avoid ill-effects in applications.

*Reproducibility.* Our methods are fairly straightforward. Most implementation details are already included in our paper. We will release our code publicly on acceptance.

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

APPENDIX

| Query Type (Train/ Eval) | Data Instance Example (Premise-Hypothesis-Label) | Structure Type | Network Size (number of nodes) |
|---|---|---|---|
| Train | Mhb causes iqB. iqB causes G. Does G cause iqB?: No | Short Linear Sequence | 3-6 |
| Train | N5w causes s. 6D causes s. Does N5w cause s?: Yes | Short Sequence with Random Flipping | 3-6 |
| Eval | w3 causes ROv. w3 causes tQC. H causes ROv. H causes tQC. b causes ROv. b causes w3. b causes H. Does tQC cause ROv?: No | Branching | 5,8,10,12 |
| Eval | LKk causes 5Ov. Kk causes L0. L0 causes KWO. 5Ov causes c. Does KWO cause L0?: No | Shuffled Sequences | 3-9 |
| Eval | FDAH26mV7 causes 7tzaIHjlY. 7tzaIHjlY causes 0kspcX95Im. 0kspcX95Im causes 7rhFSlx2o9. 7rhFSlx2o9 causes 1PlG5LHVqp. Does FDAH26mV7 cause 7tzaIHjlY?: Yes | Sequences with Longer Node Names | 3-9 |
| Eval | r causes rZ. rZ causes L. L causes bUx. bUx causes Pbr. Pbr causes 1w. 1w causes c3. c3 causes yBQ. yBQ causes yK. yK causes w. w causes P. P causes kH. kH causes 1u. 1u causes jV7. jV7 causes i. Does r cause rZ?: Yes | Long Linear Sequences | 7-15 |
| Eval | rU6 causes eF. eF causes ivC. 3R causes ivC. 3R causes A8. 2 causes A8. 2 causes i. i causes a03. y causes a03. b causes y. b causes h. h causes yN. ic0 causes yN. ic0 causes Hd. Hd causes U. Does rU6 cause eF?: Yes | Long Sequences with Random Flipping | 7-15 |

Table A1: Table with examples of data instances of different causal structural networks used for training and evaluating models. Each instance is broken down into premise, hyopthesis, and label. During evaluation, only the premise followed by the corresponding hypothesis is provided, whereas during training of transformer, the model is trained on the loss of prediction of the label token.

## A  MULTI-SHOT PROMPT

### A.1  CAUSE-EFFECT INFERENCE TASK

Chain lengths of the in context examples ranged from 3 to 6 to maintains consistency with the training and testing paradigm used for our 67-million-parameter model.

The following multi-shot prompt was used to evaluate the baselines and models across different test sets, assessing their generalization based on length, order, and branching.

*Following the given examples answer the question regarding causal relationship between two variables: '5e0 causes vAf. vAf causes VO. Does vAf cause VO?: Yes'*
*'5e0 causes vAf. vAf causes VO. Does vAf cause 5e0?: No'*
*'e0F causes Z. Z causes 0U. 0U causes mR. mR causes 1L. Does mR cause 1L?: Yes'*
*'e0F causes Z. Z causes 0U. 0U causes mR. mR causes 1L. Does Z cause e0F?: No'*
*'b causes K. K causes qPv. 5 causes qPv. Does b cause qPv?: Yes'*
*'b causes K. K causes qPv. 5 causes qPv. Does b cause 5?: No'*
*'Mhb causes t0a. 6Eh causes Mhb. NS causes 6Eh. n causes NS. n causes xu. Does xu cause 6Eh?: No'*
*'Mhb causes t0a. 6Eh causes Mhb. NS causes 6Eh. n causes NS. n causes xu. Does n cause NS?: Yes'*

## A.2 CORR2CAUSE

Below is the MultiShot Prompt for the Corr2Cause Experiment:
|SYSTEM| You are a helpful assistant on causal reasoning. Your goal is to answer factually and concisely questions about cause and effect.

|USER| Premise: Suppose there is a closed system of 4 variables, R, sG, vE and Y. All the statistical relations among these 4 variables are as follows: R correlates with vE. R correlates with Y. sG correlates with vE. sG correlates with Y. vE correlates with Y. However, R is independent of sG. Hypothesis: There exists at least one confounder (i.e., common cause) of vE and Y.

|MODEL| Yes

|USER| Premise: Suppose there is a closed system of 4 variables, uV, S, v, and pPf. All the statistical relations among these 4 variables are as follows: uV correlates with v. uV correlates with pPf. S correlates with v. S correlates with pPf. v correlates with pPf. However, uV is independent of S. Hypothesis: There exists at least one confounder (i.e., common cause) of uV and v.

|MODEL| No

|USER| Premise: Suppose there is a closed system of 3 variables, 39, 52, and fM. All the statistical relations among these 3 variables are as follows: 39 correlates with C. 52 correlates with fM. However, 39 is independent of 52. Hypothesis: There exists at least one collider (i.e., common effect) of 39 and 52.

|MODEL| Yes

|USER| Premise: Suppose there is a closed system of 3 variables, mFv, lth, and HVD. All the statistical relations among these 3 variables are as follows: mFv correlates with HVD. lth correlates with HVD. However, mFv is independent of lth. Hypothesis: There exists at least one collider (i.e., common effect) of lth and HVD.

|MODEL| No

|USER| Premise: Suppose there is a closed system of 4 variables, g1L, wlA, oO, and D. All the statistical relations among these 4 variables are as follows: g1L correlates with oO. g1L correlates with Z9. wlA correlates with oO. wlA correlates with Z9. oO correlates with Z9. However, g1L is independent of wlA. wlA and Z9 are independent given g1L and oO. Hypothesis: wlA is a cause for Z9, but not a direct one.

|MODEL| Yes

|USER| Premise: Suppose there is a closed system of 4 variables, 6na, lWS, rw, and IG. All the statistical relations among these 4 variables are as follows: 6na correlates with rw. 6na correlates with IG. lWS correlates with rw. lWS correlates with IG. rw correlates with IG. However, 6na is independent of lWS. 6na and IG are independent given lWS and rw. 6na and IG are independent given rw. lWS and IG are independent given 6na and rw. lWS and IG are independent given rw. Hypothesis: rw is a cause for lWS, but not a direct one.

|MODEL| No

|USER| Premise: Suppose there is a closed system of 3 variables, VR4, zf, and D. All the statistical relations among these 3 variables are as follows: VR4 correlates with D. zf correlates with D. However, VR4 is independent of zf. Hypothesis: zf directly causes D.

|MODEL| Yes

|USER| Premise: Suppose there is a closed system of 3 variables, uj, x, and rW. All the statistical relations among these 3 variables are as follows: uj correlates with rW. x correlates with rW. However, uj is independent of x. Hypothesis: uj directly causes x.

|MODEL| No

## B    RESULTS AND ANALYSIS

| Model | 5 | | 8 | | 10 | | 12 | |
|---|---|---|---|---|---|---|---|---|
| | BF=2 | BF=1.4 | BF=2 | BF=1.4 | BF=2 | BF=1.4 | BF=2 | BF=1.4 |
| **Baselines** | | | | | | | | |
| *Zero shot* | | | | | | | | |
| GPT-4 | 0.98 | 0.95 | 0.91 | 0.90 | 0.84 | 0.88 | 0.82 | 0.86 |
| Gemini Pro | 0.77 | 0.74 | 0.72 | 0.76 | 0.71 | 0.73 | 0.73 | 0.71 |
| Phi-3 | 0.87 | 0.83 | 0.82 | 0.79 | 0.77 | 0.77 | 0.75 | 0.80 |
| *Multi shot* | | | | | | | | |
| GPT-4 | **0.99** | **0.97** | **0.94** | **0.93** | **0.90** | **0.94** | **0.89** | **0.93** |
| Gemini Pro | 0.81 | 0.76 | 0.77 | 0.79 | 0.75 | 0.77 | 0.78 | 0.79 |
| Phi-3 | 0.77 | 0.78 | 0.79 | 0.82 | 0.78 | 0.794 | 0.80 | 0.79 |
| **Axiomatic Training** | | | | | | | | |
| OCC w NoPE | 0.52 | 0.51 | 0.53 | 0.52 | 0.52 | 0.55 | 0.49 | 0.47 |
| TS1 w LPE | 0.79 | 0.84 | 0.71 | 0.76 | 0.68 | 0.69 | 0.65 | 0.65 |
| TS1 w SPE | 0.72 | 0.79 | 0.63 | 0.64 | 0.56 | 0.61 | 0.52 | 0.59 |
| TS1 w NoPE | 0.77 | 0.84 | 0.73 | 0.76 | 0.68 | 0.70 | 0.62 | 0.66 |
| TS2 w LPE | 0.72 | 0.80 | 0.61 | 0.71 | 0.62 | 0.63 | 0.56 | 0.63 |
| TS2 w SPE | 0.52 | 0.70 | 0.49 | 0.49 | 0.49 | 0.49 | 0.51 | 0.52 |
| TS2 w NoPE | 0.83 | 0.86 | 0.74 | 0.77 | 0.69 | 0.74 | 0.64 | 0.70 |

Table A2: Evaluated on branched graphs created using Erdos Renyl, with varying branching factors (calculated by number of edges/number of nodes). TS1 and TS2 denote Pretraining data setup 1 and 2 from Section 3. OCC setup denotes Only sequential Causal Chains with no random flipping. SPE: Sinusoidal PE, LPE: Learnable PE, w/o PE: No PE. Decoder model remains the same across all setups (67 Million parameter), Accuracy metric is used. Even though the transformer was trained on linear causal chains, it still shows impressive generalization capabilities to high brnahcing factors, highlighting how data perturbations and different positional encodings aid structural generalization capabilities. Refer 4.4 for detailed result analysis.

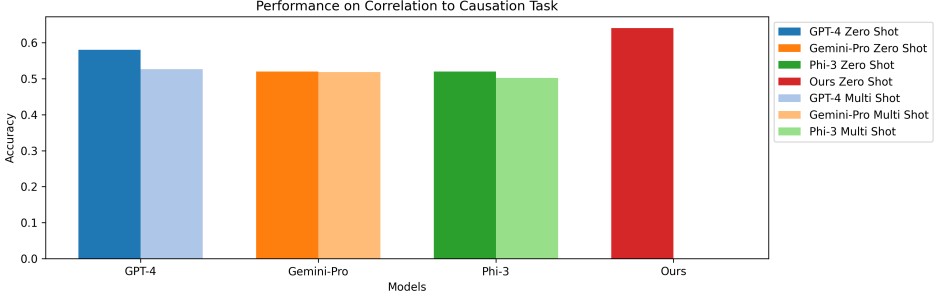

Figure A1: Correlation to Causation Experiments adapted from (Jin et al., 2024b)

| Model | 7 | | 8 | | 9 | | 10 | | 11 | | 12 | | 13 | | 14 | | 15 | |
|---|---|---|---|---|---|---|---|---|---|---|---|---|---|---|---|---|---|---|
| | FS | RF | FS | RF | FS | RF | FS | RF | FS | RF | FS | RF | FS | RF | FS | RF | FS | RF |
| **Baselines** | | | | | | | | | | | | | | | | | | |
| **Single Shot** | | | | | | | | | | | | | | | | | | |
| GPT-4 | 0.95 | 0.98 | 0.97 | 0.93 | 0.87 | 0.94 | 0.91 | 0.87 | **0.90** | 0.95 | 0.92 | 0.92 | 0.85 | 0.93 | **0.93** | 0.93 | **0.89** | 0.86 |
| Gem-Pro | 0.63 | 0.73 | 0.69 | 0.74 | 0.64 | 0.75 | 0.65 | 0.81 | 0.72 | 0.78 | 0.60 | 0.80 | 0.59 | 0.68 | 0.67 | 0.64 | 0.61 | 0.66 |
| Phi-3 | 0.81 | 0.85 | 0.96 | 0.85 | 0.85 | 0.85 | 0.87 | 0.89 | **0.90** | 0.86 | 0.84 | 0.85 | 0.91 | 0.84 | 0.90 | 0.80 | 0.78 | 0.85 |
| **Multi Shot** | | | | | | | | | | | | | | | | | | |
| GPT-4 | 0.97 | **0.99** | 0.93 | **0.99** | 0.92 | **0.96** | 0.88 | **0.94** | 0.89 | **0.97** | 0.89 | 0.93 | 0.88 | **0.95** | 0.93 | **0.94** | 0.86 | **0.94** |
| Gem-Pro | 0.80 | 0.82 | 0.81 | 0.79 | 0.78 | 0.81 | 0.67 | 0.79 | 0.73 | 0.82 | 0.74 | 0.83 | 0.67 | 0.78 | 0.72 | 0.78 | 0.68 | 0.78 |
| Phi-3 | 0.83 | 0.92 | 0.89 | 0.88 | 0.75 | 0.86 | 0.66 | 0.87 | 0.80 | 0.90 | 0.80 | 0.85 | 0.79 | 0.82 | 0.71 | 0.81 | 0.72 | 0.82 |
| **Axiomatic Training** | | | | | | | | | | | | | | | | | | |
| TS1 w NoPE | **1.00** | **0.99** | 0.95 | 0.96 | 0.88 | 0.89 | 0.76 | 0.88 | 0.73 | 0.90 | 0.77 | 0.92 | 0.61 | 0.82 | 0.67 | 0.78 | 0.68 | 0.81 |
| TS1 w LPE | 0.98 | 0.96 | 0.92 | 0.97 | 0.77 | 0.90 | 0.59 | 0.87 | 0.57 | 0.86 | 0.57 | 0.84 | 0.55 | 0.73 | 0.51 | 0.76 | 0.50 | 0.68 |
| TS1 w SPE | 0.99 | 0.95 | 0.95 | 0.94 | 0.86 | 0.76 | 0.80 | 0.75 | 0.76 | 0.79 | 0.84 | 0.68 | 0.79 | 0.63 | 0.85 | 0.65 | 0.77 | 0.69 |
| TS2 w NoPE | **1.00** | 0.98 | 0.99 | 0.97 | **0.92** | 0.91 | 0.88 | 0.90 | 0.86 | 0.92 | **0.95** | 0.90 | **0.96** | 0.83 | 0.81 | 0.84 | 0.85 | 0.78 |
| TS2 w LPE | **1.00** | 0.98 | 0.88 | 0.97 | 0.80 | 0.88 | 0.62 | 0.92 | 0.66 | 0.91 | 0.64 | 0.81 | 0.65 | 0.75 | 0.62 | 0.75 | 0.62 | 0.77 |
| TS2 w SPE | 0.95 | 0.93 | 0.81 | 0.84 | 0.56 | 0.34 | 0.50 | 0.38 | 0.50 | 0.44 | 0.51 | 0.57 | 0.46 | 0.74 | 0.52 | 0.75 | 0.50 | 0.77 |
| OCC w NoPE | 0.98 | 0.58 | 0.79 | 0.49 | 0.86 | 0.51 | **0.92** | 0.49 | 0.72 | 0.57 | 0.90 | 0.50 | 0.81 | 0.52 | 0.84 | 0.52 | 0.83 | 0.46 |

Table A3: Results on longer chains of linear sequential chains with all edges in forward direction (Only causal chains or forward sequence denoted using FS) and sequences with randomly flipped edges (Random flipping so denoted with RF). TS1 and TS2 denote Pretraining data setup 1 and 2 from Section 4. SPE: Sinusoidal PE, LPE: Learnable PE, w/o PE: No PE. Model remains the same across all setups (67 Million parameter based). For longer chains, NoPE performs best on sequential linear setup. Accuracy metric is used

| Model | 3 | 4 | 5 | 6 | 7 | 8 | 9 |
|---|---|---|---|---|---|---|---|
| **Baselines** | | | | | | | |
| **Single Shot** | | | | | | | |
| GPT-4 | **1.00** | **1.00** | **1.00** | **1.00** | **1.00** | **1.00** | **1.00** |
| Gemini Pro | 0.96 | 0.94 | 0.86 | 0.81 | 0.76 | 0.73 | 0.71 |
| Phi-3 | 0.99 | 0.98 | 0.95 | 0.94 | 0.96 | 0.95 | 0.93 |
| **Multi Shot** | | | | | | | |
| GPT-4 | **1.00** | **1.00** | 0.98 | 0.98 | 0.98 | 0.98 | 0.97 |
| Gemini Pro | **1.00** | **1.00** | 0.91 | 0.90 | 0.86 | 0.88 | 0.84 |
| Phi-3 | 0.93 | 0.89 | 0.89 | 0.84 | 0.82 | 0.77 | 0.79 |
| **Axiomatic Training** | | | | | | | |
| TS1 w NoPE | **1.00** | **1.00** | **1.00** | 0.99 | 0.99 | 0.92 | 0.91 |
| TS1 w LPE | **1.00** | 0.93 | 0.75 | 0.58 | 0.52 | 0.51 | 0.47 |
| TS1 w SPE | 0.90 | 0.79 | 0.76 | 0.70 | 0.66 | 0.73 | 0.68 |
| TS2 w NoPE | **1.00** | **1.00** | **1.00** | 0.99 | 0.97 | 0.95 | 0.95 |
| TS2 w LPE | **1.00** | 0.99 | 0.75 | 0.70 | 0.56 | 0.55 | 0.55 |
| TS2 w SPE | 0.65 | 0.50 | 0.52 | 0.50 | 0.54 | 0.48 | 0.50 |
| OCC w NoPE | **1.00** | 0.97 | 0.96 | 0.93 | 0.92 | 0.84 | 0.87 |

Table A4: Results on node name length generalization. TS1 and TS2 denote Training Data setup 1 and 2 from Section 4 **??**. OCC is the third data setup comprising of sequential causal chains. SPE: Sinusoidal PE, LPE: Learnable PE, w/o PE: No PE. Model remains the same across all setups (67 Million parameter based). For longer node names, NoPE performs best on sequential linear setup. Accuracy metric is used.

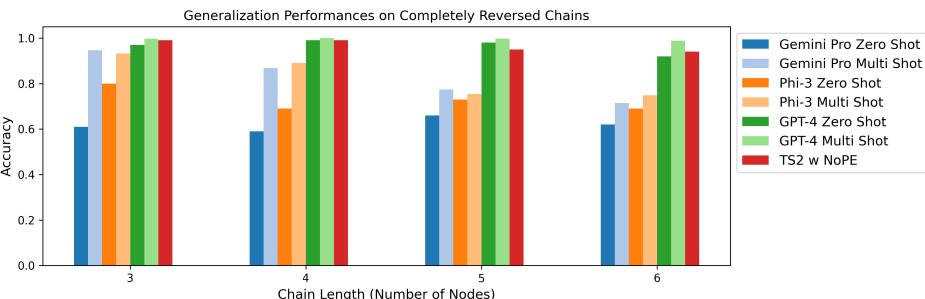

Figure A2: Performance comparison of our best performing transformer model trained on TS2 with NoPE (trained without any completely reversed chains), against larger models like GPT-4, Gemini Pro, and Phi-3.

## C    PRELIMINARIES AND NOTATIONS

**Causal Models**    Let $\mathcal{M} = (\boldsymbol{X}, \boldsymbol{U}, \mathcal{F})$ be a causal model defined over a set of endogenous variables $\boldsymbol{X}$, exogenous variables $\boldsymbol{U}$ and the causal relationship between then defined by set of structural equations $\mathcal{F}$ (Galles & Pearl, 1997). Let $\mathcal{G}$ be the causal graph associated with the causal model $\mathcal{M}$ where the nodes $\boldsymbol{V}$ in $\mathcal{G}$ correspond to the variables in $\mathcal{M}$ and an edge $V_i \to V_j$ between any two nodes $V_i, V_j$ denote the causal relationship between them. The causal relationship of node $X_i$ is characterized by the functional relationship $f_i \in \mathcal{F}$ s.t., $x_i = f_i(\boldsymbol{pa}_i, \boldsymbol{u}_i)$. Here $\boldsymbol{pa}_i$ are the parent of the node $X_i$ is the corresponding causal graph $\mathcal{G}$ and $\boldsymbol{u_i} \subseteq \boldsymbol{U}$ are set of exogenous variables influencing the exogenous variable $X_i$. In our work, we assume that there are no hidden confounders so we have one exogenous variable corresponding to every endogenous variable i.e. $\boldsymbol{u}_i = u_i$. Each exogenous variable has an associated probability distribution which quantifies the uncertainty in the system i.e. $u_i \sim \mathbb{P}(u_i)$. Thus the joint distribution of the exogenous variable is given by $\mathbb{P}(\boldsymbol{U})$. Since any endogenous variable is a deterministic function of other endogenous and exogenous variables the probability distribution corresponding to the endogenous variable is the push-forward of the exogenous variable i.e $\mathbb{P}(\boldsymbol{X}) \triangleq \mathbb{P}^{\#}(\boldsymbol{U})$.

**d-separation**    The causal graph $\mathcal{G}$ encodes the conditional independence in the corresponding probability distribution using path separations statements called *d-separation*. Two sets of random variable $\boldsymbol{X}_i$ and $\boldsymbol{X}_j$ are conditionally independent of each other conditioned on $\boldsymbol{X}_z$ if all the undirected path between $\boldsymbol{X}_i$ and $\boldsymbol{X}_j$ in $\mathcal{G}$ are blocked by $\boldsymbol{X}_z$. A path between $\boldsymbol{X}_i$ and $\boldsymbol{X}_j$ is blocked if there exists a node $A \in \boldsymbol{X}_z$ s.t. it satisfies one of the following conditions: 1. A is a common cause in the path (i.e. $\cdot \leftarrow A \to \cdot$), 2. A is a mediator in the path (i.e. $\cdot \to A \to \cdot$), or 3. A is not a collider (i.e. $\cdot \to A \leftarrow \cdot$) or descendent of any collider in the path (see Pearl (2009a) for details.)

### C.1    DEFINITIONS

Following the formal definitions provided by (Jin et al., 2024b), we explain the following terminologies:

**Markov Property** In a directed acyclic graph (DAG) $G$, the Markov property asserts that each node $X_i$ is conditionally independent of its non-descendants given its parents. This can be written as $X_i \perp\!\!\!\perp \text{NonDe}(X_i) \,|\, \text{Pa}(X_i)$, where $\text{NonDe}(X_i)$ represents the set of non-descendants of $X_i$, excluding the node itself, and $\text{Pa}(X_i)$ denotes its parents. Leveraging the Markov property, the joint distribution over all the nodes can be factorized as:

$$P(X_1, \ldots, X_N) = \prod_{i=1}^{N} P(X_i \,|\, \text{Pa}(X_i)).$$

**Markov Equivalence Class** Two directed acyclic graphs (DAGs) are considered Markov equivalent if they induce the same joint distribution $P(X)$. The collection of DAGs that are Markov equivalent

is referred to as a Markov equivalence class (MEC). Causal graphs within the same MEC can be easily recognized as they share the same skeleton (i.e., undirected edges) and V-structures (i.e., configurations of the form $A \rightarrow B \leftarrow C$, where $A$ and $C$ are not directly connected).

## D  POSITIONAL ENCODINGS AND THEIR ROLE IN GENERALIZATION

Positional Encoding (PE) play a crucial role of providing information about the absolute and relative position of tokens in a sequence (Vaswani et al., 2023). (Vaswani et al., 2023) propose an absolute positional encoding strategy using periodic functions (e.g., sinusoidal or cosine) to initialize these encodings. Absolute positional encoding provides definite values for all positions across any sequence length. However, studies (Ontañón et al., 2022; Csordás et al., 2021) show absolute positional encoding fails in length generalization tasks for transformers. In the learnable APE variant (Radford et al., 2018), each positional embedding is randomly initialized and trained with the model. This approach falters with sequences longer than those seen in training, as the new positional embeddings remain untrained and randomized. Interestingly, recent findings (Kazemnejad et al., 2023; Haviv et al., 2022) indicate that removal of PEs in auto-regressive models can improve model's length generalization capabilities, wherein the attention mechanism during auto-regressive decoding is sufficient to encode positional information.

## E  FORMALISING TRAINING AND EVALUATION SETUP

Let $f_{dim}$ represent the maximum value for a given perturbation dimension $dim$, along which we construct train and evaluation sets for our axiomatic framework. For each dimension, we choose a threshold $\tau_{dim} \in L$, such that $f_{dim} < \tau_{dim}$ forms our training set and $f_{dim} \geq \tau_{dim}$ forms the evaluation set. So, $f_{dim} \in \{f_{len}, f_{branch}, f_{nodelen}, f_{revfactor}, f_{shuffle}\}$ where:

- $f_{len} = \max_{\forall i}(len(V_i))$, gives the maximum number of nodes across all causal sequences. $\tau_{len}$ for length is set at 6, with $f_{len} \in [3, 6]$.
- $f_{branch} = \max_{\forall i}(|X_i|/|V_i|)$ gives the maximum branching factor in a dataset, with $\tau_{branch} = 0.8$ (for 6 node linear sequences). For sequences in the train set, the branching factor ranges from 0.6 to 0.8 for 3 to 6 length sequences.
- Let $l_{i,j}$ be the length of the name of the node $X_{i,j}$, then $l_{i,j} = (len(X_{i,j}))$.Therefore, the maximum length of node names across all nodes in all causal sequences can be represented as: $f_{nodenamelen} = \max_{1 \leq i \leq n,\, 1 \leq j \leq m} l_{i,j}$. We set $\tau_{nodelen}$ for train set as 3, with $f_{nodelen} \in [1, 3]$.
- Given any causal sequence $X_i$ and a function $N$, where $N(X_{i,j}, X_{i,j+1})$ returns natural language representation of a directed edge between $j$ and $j + 1$ node in the causal chain $X_i$. $f_{shuffle} = \cap_{\forall i,j} \text{Perm}(N(X_{i,j}, X_{i,j+1}))$, where $N(X_{i,j}, X_{i,j+1})$ represents deviation from original sequential order of natural language sentences to represent $X_i$.
- Given a causal sequence $X_i$ and let $R(X_i, f_{revfactor})$ be an operation on the causal chain that flips the direction of every edge in the sequence with probability $f_{revfactor}$. In the training set, there is a directed edge between every sequential pair of nodes $X_{i,j}, X_{i,j+1}$ with $f_{revfactor} = 0$ (for linear sequence, $X_{i,j} \rightarrow X_{i,j+1}$) or 0.5 (for sequence with random flipping, $X_{i,j} \rightarrow X_{i,j+1}$ or $X_{i,j} \leftarrow X_{i,j+1}$) In the evaluation set $f_{revfactor} = 1$ i.e., all sequences for reversal evaluation setup are completely reversed unlike in train set where no sequence is present where all edges are completely reversed.

