# OpenReview forum: "Teaching Transformers Causal Reasoning through Axiomatic Training"
_ICLR.cc/2025/Conference — Submitted to ICLR 2025_

### Official Review · Reviewer_kNHW · 2024-10-22

**Soundness:** 2
**Presentation:** 3
**Contribution:** 2
**Rating:** 3
**Confidence:** 3

**Summary:**

The paper mainly presents a synthetic data generation protocol for axiom learning in transformers. Through training a GPT-2 like transformer on the synthetic demonstrations, experiments show good generalization ability when evaluating it on a larger network. The paper is well formatted and the axiom generation procedure is well explained.

**Strengths:**

1. Teaching LLM to reason is an important topic.
2. Using high-quality synthetic data for pertaining LLM is important and the explanations of the synthetic axiom generation are clear.

**Weaknesses:**

1. There is no proper explanation for why training the transformer on the given symbolic expressions will generalize better and even outperform the SoTA GPT-4 model. If my understanding is correct, the author only changes the dataset for pertaining and still uses negative log-likelihood (SFT) as the loss function to train the transformer.  Is it possible the model overfits the symbolic expression dataset the author has provided?
2. The author attempts to investigate how different embedding techniques, i.e., SPE, LPE, and NoPE, will affect the causal reasoning ability of LLM. As far as I know, most open-source LLMs like llama, they are all using RoPE [1] for positional embedding which is a kind of relative embedding suitable for format learning. I am curious why the author did not mention RoPE in this paper. Moreover, the evaluation of positional encoding is only limited to the final performance and I could not find any deep insight into why SPE will perform better. [2] is a good reference paper for probing different LLM layers and testing their effectiveness.

[1] Jianlin Su et al. RoFormer: Enhanced Transformer with Rotary Position Embedding

[2] Tian Ye et al. Physics of Language Models: Part 2.1, Grade-School Math and the Hidden Reasoning Process

**Questions:**

There are some minor questions that need further clarification
1. Line 317 "We train a decoder-based 67 million parameter model based on GPT-2’s architecture". Does it mean the model in the paper is trained from scratch?

---

> ### Author Response · Authors · 2024-12-04
> **Response to Reviewer kNHW**
>
> > There is no proper explanation for why training the transformer on the given symbolic expressions will generalize better and even outperform the SoTA GPT-4 model. If my understanding is correct, the author only changes the dataset for pertaining and still uses negative log-likelihood (SFT) as the loss function to train the transformer. Is it possible the model overfits the symbolic expression dataset the author has provided?
>
> The generalization capabilities of models for applying causal axioms on unseen and larger causal graphs come from two main factors: structured perturbations in synthetic training data and changes in model’s architecture (applying/removing positional encodings). While the training data consists of simplistic linear causal chains represented in natural language, the evaluation set consists of much more complex causal graphs in terms of branching, length, size and even order. Therefore, if the model had fitted over the causal sequences in the training data made of small linear causal chains, it wouldn’t have been able to perform well on unseen and much complex causal networks that are present in the evaluation set. Overfitting on training instances would have led to a drop in performance when it encountered longer, more branched or completely reversed chains during evaluation, since it had not seen any of such instances during its training.
>
> Our contribution is not just limited to training the model on synthetic data from scratch, but investigating what perturbations in training data aid generalization the most, along with what architectural changes and model training decisions (like removal of Positional encodings), helps the model perform well on OOD evaluation set.
>
> > The author attempts to investigate how different embedding techniques, i.e., SPE, LPE, and NoPE, will affect the causal reasoning ability of LLM. As far as I know, most open-source LLMs like llama, they are all using RoPE [1] for positional embedding which is a kind of relative embedding suitable for format learning. I am curious why the author did not mention RoPE in this paper. Moreover, the evaluation of positional encoding is only limited to the final performance and I could not find any deep insight into why SPE will perform better. [2] is a good reference paper for probing different LLM layers and testing their effectiveness.
>
> We thank the reviewer for this point. Past literature evaluating impact of positional encodings on length generalization focused on the positional encodings we had evaluated on, therefore we decided to extend on that direction and verify whether the findings remain consistent when the evaluation goes beyond typical length generalization (for example, evaluating on more branching, longer node names, etc). We can incorporate RoPE as well in our evaluation.
>
> On why SPE performs comparatively well on some evaluation benchmarks, SPE was introduced as an alternative of LPE to tackle the problem of length generalization for transformers. Since LPE requires training positional encoding like a typical ML parameter, the PE for longer sequences will be randomized, thus affecting the model’s performance. In comparison, SPE follows a sinusoidal function to get the value for each position, therefore removing the need to train positional parameters. This aims at improving length generalization, and our results corroborate this. However, with SPE, we also observe surprising performance drops on some evaluation sets. Here’s an explanation: since SPE has a component of sinusoidal function as part of the positional encoding, this introduces an aspect of periodicity. Our synthetic training instances also follow a periodic pattern (“A causes B. B causes C. C causes D….”). Since the model is trained on such recurring periodic training instances, we believe the model picks up spurious correlations based on that. This is also why we see a surprising performance drop when the length of the node names is increased, hindering the periodic flow of each instance.
>
> > Line 317 "We train a decoder-based 67 million parameter model based on GPT-2’s architecture". Does it mean the model in the paper is trained from scratch?
>
> Yes, the model is trained from scratch on our synthetic training data consisting of linear causal sequences.

---

### Official Review · Reviewer_zeDh · 2024-11-02

**Soundness:** 2
**Presentation:** 2
**Contribution:** 2
**Rating:** 5
**Confidence:** 3

**Summary:**

The paper investigates the problem of causal reasoning in transformer models. Specifically, authors try to come up with a different training protocol for enabling causal reasoning, which they call axiomatic training. The dataset for axiomatic training consists of a large number of demonstrations of (abstract) causal reasoning - in this work, primarily based on one specific axiom, transitivity. One such demonstration could look like "X causes Y, Z causes Y. Does Z cause X?" followed by an answer "No". After generating a large dataset of those types of prompts, varying lengths, variable names and causal graphs topology, a small transformer model is trained to predict the last token on different subsets of the dataset, to check what levels of generalisation it would reach.
Trained model is compared with off-the-shelf LLMs (GPT-4, Phi-3, Gemini). Empirical comparison is extended further by looking at more general correlational statements from Corr2Cause dataset. The paper also investigates the impact of positional encoding on generalisation abilities. Given the results, authors suggest that similar techniques might inspire specific ways of synthetic data augmentation and help with LLM reasoning.

**Strengths:**

- The problem of reasoning, and, specifically, causal reasoning, is important and not very well understood yet, which makes the paper topical and interesting.
- The axiomatic training method suggested in the paper seems a priori reasonable - LLMs get access to many demonstrations of an axiom, but  augmenting the data in specific ways might yield better results. The fact that the model performs better, or on par, with off-the-shelf, cutting-edge LLMs is evidence that the method has some potential to improve training.
- Empirical testing of various levels of generalisation for the transitivity axiom seems solidly done, including retraining the model on a filtered data to test specific claims.

**Weaknesses:**

- Writing was a bit disjointed. The main thread of the narrative was to investigate generalisation of a specific training setup. But then there was also some (disjointed) discussion of positional encoding, which - although important and informative - was distracting from the main story. Some other pieces of writing were also a bit confusing, such as the transitions between sections 3.2, 3.3 and 4.1 (e.g. random flipping is described twice in 3.2 and 3.3, then branching is described twice in 3.3 and 4.1) - it might be good to consolidate those sections.
 - Constituting most of the paper - investigation into transitivity axiom alone seems underwhelming, in relation to the claims about teaching transformers causal reasoning. Transitivity check does not involve any substantial *causal* component - it is simply a question about graph reachability.
 - Thus, the most interesting section, and the only one that truly uses the heavy machinery of d-separation etc., is Section 5. But it is very short, and the full details are not even included in the appendix. Section starts by describing the task in the benchmark as "significantly harder" because of various potential questions, but this feels misleading, since the full benchmark is not used. There is no formal definition of what the "correlational statement" is, which that empirical evaluation measures - appendix A.2 simply lists all possible questions.
- The task is also quite underwhelming on a conceptual level - since the variables names are repeated, it is not clear to me to what extent the model could have just memorised some patterns, which are enough to get to 64% accuracy, as the data sizes are very small - only graphs with 3 or 4 nodes (respectively 6, 31 non-isomorphic DAGs) and testing on 5 nodes (302 non-isomorphic DAGs).
- It would be good to include a theoretical discussion as to whether we can expect transformer-based models to solve this task in principle - I don't know if the algorithm of checking d-separation can be implemented in a single pass of a bounded-depth circuit.
- In general, this sort of technique seems to quickly hit diminishing returns, as per the bitter lesson. It is not a general-purpose algorithm, as it requires very specific data augmentation for different logic and reasoning systems. I would be more interested in leaning deeper into trying to infer generalisation properties of the architecture.

**Questions:**

Questions:
- Recent research into decision transformers showed their significant issues with trajectory stitching (see e.g. Zhou et al. 2024, "Free from Bellman Completeness"). Do you think it has any relation to your problem of computing graph reachability?
 - Why do you think the model truly generalised in Sec 5, and not just remembered some patterns, given the very effective dataset size?
 - What is the definition of the correlational statement, and what is the complexity of the algorithm that answers it?
 - Do you think tokenizer issues might have impacted off-the-shelf LLMs abilities to perform on the task?

Small things:
 - Citations in line 347 and label in line 914 are not displayed correctly
 - Colours are inverted in Fig A1.

---

> ### Author Response · Authors · 2024-12-04
> **Response to Reviewer zeDh**
>
> > Writing was a bit disjointed. The main thread of the narrative was to investigate generalisation of a specific training setup. But then there was also some (disjointed) discussion of positional encoding, which - although important and informative - was distracting from the main story.
>
> We thank the reviewer for this point. To tackle the first point on Positional Encodings (PEs), multiple papers have extensively evaluated and showed how positional encodings play a crucial role in aiding language models' length generalization capabilities. An interesting finding from this literature is that NoPE is an effective way of generalizing to longer lengths than those in the instances in the training set (for decoder-based models). Even in our evaluation setting, assessing structural generalization across different dimensional aspects (like length, branching, order), we observe how models' capabilities vary with different positional encodings. Therefore, we highlight the variation with positional encoding while explaining our training recipe for axiomatic training. We will make sure to improve the flow and make this easily understandable to remove the problem of disjointed discussion.
>
> As for the second point, in section 3.2 we try to define the training setup and the different structural perturbations we did in the training instances to aid structural generalization.  In section 3.3 however, we explained how our evaluation set was constructed, and the different levels of complexities we added to make the task as hard as possible for the transformer. We will incorporate these points and reduce redundancy in the final draft by consolidating the sections as suggested.
>
> > Constituting most of the paper - investigation into transitivity axiom alone seems underwhelming, in relation to the claims about teaching transformers causal reasoning. Transitivity check does not involve any substantial causal component - it is simply a question about graph reachability.
>
> Our axiomatic setting and evaluation offer an initial step to understand how transformers could be potentially trained for rules/axioms that could help solve causal problems. Since causal reasoning can be broken down into axioms, we started with the transitivity axiom. Overall, the ability to perform accurate graphical operations (like traversal) can be highly impactful while dealing with causal graphs. As we mention in the discussion, the same kind of training can also be applied to other symbolic reasoning systems, such as logical reasoning.
>
> > Thus, the most interesting section, and the only one that truly uses the heavy machinery of d-separation etc., is Section 5. But it is very short, and the full details are not even included in the appendix. Section starts by describing the task in the benchmark as "significantly harder" because of various potential questions, but this feels misleading, since the full benchmark is not used. There is no formal definition of what the "correlational statement" is, which that empirical evaluation measures - appendix A.2 simply lists all possible questions.
>
> We used a smaller section of the benchmark, more specifically correlational statements of smaller graphs to train our model through the axiomatic framework. We evaluated how well the model performed on statements of much larger networks which were not a part of the training setup. We believe this is a challenging problem, since generalization to larger unseen networks can be a good indicator of transformers’ ability for learning such tasks.
> We will try to provide a more detailed explanation of correlational statements and its formal definition.
>
> > The task is also quite underwhelming on a conceptual level - since the variables names are repeated, it is not clear to me to what extent the model could have just memorised some patterns, which are enough to get to 64% accuracy, as the data sizes are very small - only graphs with 3 or 4 nodes (respectively 6, 31 non-isomorphic DAGs) and testing on 5 nodes (302 non-isomorphic DAGs).
>
> While the variable names can be memorized, their relationships are randomized. Further, we expect that 5-length graphs will offer new structures that have never been seen in training. However, we do agree that evaluating on a larger test set will be more informative.

---

> > ### Author Response · Authors · 2024-12-04
> > **(Contd.) Response to Reviewer zeDh**
> >
> > > It would be good to include a theoretical discussion as to whether we can expect transformer-based models to solve this task in principle - I don't know if the algorithm of checking d-separation can be implemented in a single pass of a bounded-depth circuit.
> >
> > We thank the reviewer for raising this important question. We agree with the intuition of the reviewer that with a bounded depth circuit transformer it might not be possible to identify all the d-separation relations since one might need to perform an exponential number of independence tests. However, if our goal is to identify the causal graph from such d-separating sets, there have been some recent works from Shiragur et al. 2024 [1] where they aim to learn a coarser representation of the causal graph using a polynomial (in the number of variables) number of independence tests. We conjecture that it may be possible to learn such a coarser representation using bounded-depth transformers.
> >
> > > In general, this sort of technique seems to quickly hit diminishing returns, as per the bitter lesson. It is not a general-purpose algorithm, as it requires very specific data augmentation for different logic and reasoning systems. I would be more interested in leaning deeper into trying to infer generalisation properties of the architecture.
> >
> > While it might be true that scale can outperform such techniques, our analysis of billion scale models like Gemini Pro and Phi-3 reflect surprising performance failure on our synthetic evaluation tasks. We aim to provide a robust training recipe to incorporate such axiomatic understanding as an inductive bias in language models to overcome such drawbacks. We also believe such techniques can help smaller models become powerful reasoning agents and overcome possible spurious correlations picked up during training. While scale can possibly bridge the current gap, the possibility of smaller models becoming better at applying such axioms accurately with a simplistic training setup made of synthetic data can have strong implications.
> >
> > > Recent research into decision transformers showed their significant issues with trajectory stitching (see e.g. Zhou et al. 2024, "Free from Bellman Completeness"). Do you think it has any relation to your problem of computing graph reachability?
> >
> > We thank the reviewer for bringing this connection to our attention. The transitivity axiom for causal relevance (wherever applicable) can be thought of as equivalent to graph reachability and thus will bear resemblance to other problems in other domains like Reinforcement Learning (RL). We would like to point out one distinction of our problem from the paper mentioned in the comment – In RL, the goal of the agent is to explore the state space of the system which could be exponentially large in the number of variables associated with the system and thus might not scale or be learnable with finite depth transformers. However, our task (for transitivity axiom) is simpler where the goal is to see whether the random variables in the associated causal graph are connected or not. Since poly-time algorithms in the number of nodes exist for finding graph reachability, we conjecture that the transformer should be able to learn even with finite depth.
> > That said, the transitivity axiom is one of the axioms of causal irrelevance and our framework allows for other axioms to be incorporated where training the transformer might not resemble the graph reachability problem.
> >
> > > Do you think tokenizer issues might have impacted off-the-shelf LLMs abilities to perform on the task?
> >
> > There is a slight possibility that tokenizers have an impact on the LLM abilities, although if that was the case then it should have affected all evaluating settings. While Gemini Pro and Phi-3 perform fairly well across linear chains, they tend to show failure modes on setups with more branching or different order. This indicates that there may a more fundamental issue at play than just tokenization.

---

### Official Review · Reviewer_RpKQ · 2024-11-03

**Soundness:** 3
**Presentation:** 3
**Contribution:** 3
**Rating:** 6
**Confidence:** 3

**Summary:**

This paper introduces an "axiomatic training" framework for teaching transformers causal reasoning through symbolic demonstrations of causal axioms. The key contributions are:

1. A method to generate synthetic training data containing demonstrations of causal axioms
2. Empirical evidence that a 67M parameter transformer trained on simple causal chains can generalize to more complex causal structures
3. Application to both transitive causal reasoning and inferring causation from correlation statements
4. The authors demonstrate that their model often outperforms larger language models like GPT-4 and Gemini Pro on these causal reasoning tasks.

**Strengths:**

1. Novel framework for teaching causal reasoning through symbolic demonstrations
2. Comprehensive evaluation across multiple generalization dimensions
3. Strong empirical results showing competitive performance with much larger models
4. Clear potential for extending to other axioms and reasoning tasks

**Weaknesses:**

1. Experimental Design Limitations:
- The fixed training regime of 100 epochs lacks justification and sensitivity analysis
- Insufficient ablation studies on model size impact
- Limited exploration of different positional encoding schemes' effects on performance
2. Baseline Comparison Issues:
- The baseline methods may not adequately handle noise and "synthetic data"
- Potential fairness concerns in comparisons due to the presence of contradictory causal relationships in training data
- The baseline methods might need adjustment to better align with the causal reasoning tasks
3. Semantic Consideration:
- Insufficient attention to the role of semantic information in causal reasoning tasks
- Limited exploration of how semantic understanding could enhance model performance
- Potential unexplored opportunities in leveraging Transformer's semantic capabilities

**Questions:**

1. Could you provide sensitivity analyses for:
- Training epochs
- Model size variations
- Different positional encoding schemes
2. How do you ensure fair comparison with baselines given the synthetic nature of the training data?
3. Have you considered incorporating semantic understanding mechanisms to enhance the model's causal reasoning capabilities?

---

> ### Author Response · Authors · 2024-12-04
> **Response to reviewer RpKQ**
>
> > Could you provide sensitivity analyses for: Training epochs, Model size variations, Different positional encoding schemes
>
> Our earlier analysis showed no major performance difference when working with different number of epochs (ranging from 100 to 300), therefore we decided to go ahead with 100 to optimize on training time while ensuring good generalization performance. We provide analysis of popular positional encodings and its impact on generalization for our task (refer Table 1, A2, A3, A4 for results on using different positional encodings on different structural generalization). We further dive into its analysis in Section 4.5 of the paper.
>
> > How do you ensure fair comparison with baselines given the synthetic nature of the training data?
>
> We provide two types of evaluation setups while comparing with baseline LLMs, zero and multi-shot. While in zero shot we just explain the task, show the premise and ask the hypothesis directly, multi-shot setting shows multiple training instances as a part of the input prompt thus providing better context about the task. Multiple pre-existing works show how in-context examples lead to significant performance gains for LLMs. Our results corroborate this, as we see how incorporating example instances lead to performance gains for the task as compared to zero shot setting (refer tables 1, 2, A2, A3, A4). In particular, GPT-4 in multi shot setting is comfortably able to outperform all other methods, including ours across all the evaluation sets we work with showcasing how providing synthetic examples significantly improve the models’ performance.
>
>
> > Have you considered incorporating semantic understanding mechanisms to enhance the model's causal reasoning capabilities?
>
> We feel this is one of the most exciting and natural next steps of our work. We talk briefly about this in future directions in the paper and strongly believe semantic understanding can further enhance the model's utility for causal reasoning tasks for real world entities.
>
> **Justification for the Synthetic task.** While the task is synthetic in nature, this allows us to control for issues like data contamination, allowing us to truly assess models' causal reasoning and causal graph traversal capabilities, and not worry about potential memorization problems. Also, even though the evaluation setting is purely synthetic, it is still a relatively easy task focusing on small to medium scale graphs with perturbations at different scales. Given that current billion scale models are able to solve many complex problems (mathematical reasoning, code generation, and more), the failure mode on such simple evaluation sets provide interesting insights on room for improvement for such models.
>
> Specifically, while past works (Kiciman et al. 2023, Ban et al., 2024) focus on utilizing LLMs factual knowledge and semantic understanding for the task of causal discovery, it becomes difficult to disambiguate whether this performance is due to true understanding or because of potential memorization. Inspired by Corr2Cause (Jin et al. 2023) and Cladder (Jin et al. 2023) that propose synthetic benchmarks for causal inference tasks and evaluate LLMs performance on them, we wanted to understand whether we could teach a small-scale transformer the rules required to solve such tasks. Synthetic setup was the first natural step in this direction as it allowed us to accurately evaluate whether the model is learning and not depending on correlations picked up during its training for its semantic understanding.

---

### Official Review · Reviewer_KcTN · 2024-11-04

**Soundness:** 3
**Presentation:** 2
**Contribution:** 2
**Rating:** 3
**Confidence:** 5

**Summary:**

This paper introduces a way of training Transformers from scratch to learn causal reasoning axioms. The idea, called axiomatic training, is to demonstrate axioms through labeled examples that encode whether or not a hypothesis is true given a premise. For example, the premise can be conditional independences entailed by a causal graphical model, the hypothesis, whether or not some variable is a collider. To study axiomatic training, the paper considers the transitivity of causal relationships as an axiom and develops an extensive experimental design that evaluates various training strategies on different forms of generalization at test time. The paper consistently finds that their trained Transformers outperform many LLMs, while performing comparably to GPT-4. As a second application, they use axiomatic training to study performance on queries about additional graphical relationships such as colliders, direct or indirect causes and find that axiomatic training again generalized favorably compared to several LLMs.

**Strengths:**

+ With the exception of a few expositional sentences, the paper is clear about definitions, training, and the experimental setup.

+ To the best of my knowledge, the paper considers a novel setup for training Transformers to apply logical rules, at least in the context of causal reasoning.

+ The empirical setup is thorough, identifying four ways of generating out-of-distribution premises at inference time and teasing apart the impact of multiple aspects of the training setup.

**Weaknesses:**

+ At the highest level, it would be useful to get more perspective on the significance of training Transformers from scratch to automatically learn to resolve logical causal statements. First, the empirical findings suggest that GPT-4 is pre-trained to handle these queries in generalizable way already. Second, even if axiomatic training were improved to outcompete GPT-4 on these structured queries, how can this technique be incorporated into the larger decision making context? Much of text-based decision making won't feature explicit premises like conditional independence statements but rather informal natural language.  Relatedly, although the paper does summarize related work, the authors can better contextualize the significance of their contribution in the broader context of LLM-based causal reasoning. They discuss some papers, and use the Corr2Cause benchmark -- in what way does this training procedure build on the promise of LLMs answering causal questions factually?

+ I found some sentences hard to process, potentially because they were using jargon and could have been put more simply. For example, in lines 44-45, in the context of causal models, the data-generating process arguably entail the axioms -- it's not that the axioms of d-separation entail a causal model -- and so the phrase "data that is result of axioms obeyed by a data-generating process" is hard to digest. In section 3 in line 131, it would have been useful if phrases like "causal principles" were explained better before being invoked. In lines 134-135, "finite set of axioms ... completely characterized by axioms of path interception ..." is again hard to parse.

+ Relatedly, it's hard to link the transitivity axiom in (1) back to the logical formula given above it -- is the transitivity axiom supposed to be one of the six axioms you reference?

+ Some technical details are missing. In assumption 3.1, $\mathcal{M}$ is conventionally used to define a structural causal model (SCM). Is that what you're referring to? If so, the paper ought to define a (probabilistic) SCM, and clarify phrases like "all possible probability distribution(s) over $\mathcal{M}$", which again don't parse. In section 5, when applying axiomatic training to identify additional graphical relationships, it's not clear how the examples are created since many graphical relationships (e.g., direct causation) are not identified from conditional independence statements. Thus, for many premises and hypothesis pairs, there is no way to arrive at a yes or no label. How is this handled?

+ Section 3 feels like it is both missing some details (see above) but also introducing a lot of notation without giving context. In particular, the build up of the conjunctive normal form culminates in Equation (1), which is never explained simply in plain English. But looking at it, it seems to just say that (indirect) causation is transitive. There is a big notation overhead to arrive at a fairly simple idea, which makes the lead-up feel like it's purposefully trying to create an air of complication that's not needed.

**Questions:**

+ Could the authors contextualize the uses for axiomatic training in the broader causal inference and decision making context to help shed light on its significance as a training technique?

+ Can you clarify if the transitivity axiom in (1) is linked to the previous logical rules by being one of the six axioms you reference?

+ Can you clarify how examples are generated for section 5, given that many graphical relations in a causal model are not identified by conditional independences alone?

+ Can you provide some more plain English context about the transitivity axiom and its importance? It seems like it's a statement about the transitivity of indirect causation, where causation is codified through causal (ir)relevance. Is that fair to say?

---

> ### Author Response · Authors · 2024-12-04
> **Response to Reviewer KcTN**
>
> > At the highest level, it would be useful to get more perspective on the significance of training Transformers from scratch to automatically learn to resolve logical causal statements. First, the empirical findings suggest that GPT-4 is pre-trained to handle these queries in generalizable way already. Second, even if axiomatic training were improved to outcompete GPT-4 on these structured queries, how can this technique be incorporated into the larger decision making context? Much of text-based decision making won't feature explicit premises like conditional independence statements...
>
> We thank the reviewer for this point. Our axiomatic setting can be beneficial for both symbolic as well as informal text-based reasoning problems, since many such reasoning problems involve logical or causal axiomatic understanding. In addition, such a model can be used as a verifier at inference time for LLMs such as GPT-4, ensuring that the output conforms to causal axioms. We also talk about this in the “Discussion and Conclusion” section.  Thus, for formal reasoning problems, where the premise provides explicit causal or correlational relationships in natural language, a language model trained on axioms represented through symbolic text can be directly used for evaluating the hypothesis.
>
> As for reasoning-based tasks in informal natural language, an interesting direction where axiomatic training could be useful is to incorporate causal/logical axioms as inductive biases, which in turn helps in solving the end problem.  For example, given deductive reasoning problems (such as in ProtonQA (Saparov and He, 2023) , Folio (Han et al., 2024)) , which requires understanding of transitivity to identify the relationship between different entities, a model having an inductive bias of the transitivity axiom might be better equipped to solve such tasks, as compared to a model which is not able to perform well on such problems. More generally, as LLMs are known to have various reasoning failures, axiomatic training can help language models, especially smaller language models, incorporate causal or logical axioms and in turn improve reasoning performance.
>
> > Relatedly, it's hard to link the transitivity axiom in (1) back to the logical formula given above it -- is the transitivity axiom supposed to be one of the six axioms you reference?
>
> Galles and Pearl (1997) [1] axiomatize the causal irrelevance with 6 axioms. Among them is the axiom of transitivity (also restated in Equation 1, Line 170) which states that two variables are causal irrelevant if no path is connecting them in the corresponding causal graph. To simplify the training process, we take the contrapositive statement of this axiom and convert the causal irrelevance statement into a statement of causal relevance (stated in Line 172). This causal relevance statement doesn't hold in general for any probabilistic causal models. However, following Sadeghi and Soo (2024) [2], in our work we only consider the causal models for which transitivity holds for causal relevance.
>
> > Some technical details are missing. In assumption 3.1, M is conventionally used to define a structural causal model (SCM). Is that what you're referring to? If so, the paper ought to define a (probabilistic) SCM, and clarify phrases like "all possible probability distribution(s) over M"... In section 5, ... it's not clear how the examples are created since many graphical relationships (e.g., direct causation) are not identified from conditional independence statements. Thus, for many premises and hypothesis pairs, there is no way to arrive at a yes or no label. How is this handled?
>
> Yes, with “M” we refer to the skeleton of the causal graph following the notation from Galles and Pearl 1997 [1]. We thank the reviewer for pointing this out and we will include the definition of probabilistic SCM. Given a causal model “M”, the probabilistic causal model is defined by a pair <M,P(U)> where P(U) is the probability distribution over all the exogenous variables which defines the probability distribution over the endogenous variables i.e. P(V).
> As for identifying graphical relationships in section 5, we use a subset of the dataset from the Corr2Cause dataset [Jin et al. (2023)]. They included only those questions that can be answered given the input statements.
>
> > Section 3 feels like it is both missing some details (see above) but also introducing a lot of notation without giving context. In particular, the build-up of the conjunctive normal form culminates in Equation (1), which is never explained simply in plain English...There is a big notation overhead to arrive at a fairly simple idea...
>
> We thank the reviewer for pointing this out. We have provided the explicit axioms as stated in the original work from Pearl and Galles 1997 for completeness. We agree with the reviewer about the notational overhead and we will improve the notation and add other missing definitions in the revised version of the paper.

---

### Meta-Review · Area_Chair_uhq7 · 2024-12-24

**Metareview:**

The paper investigates the ability of transformers to learn the axioms of causal reasoning.  Instead of directly structuring a model to follow the axioms of causal reasoning, the model is trained with data to learn the axioms.  The strengths of the paper are the interesting investigation and good write up.  The weaknesses include limited experimentation with general axioms of causality beyond transitivity and data that is restricted to abstract symbols instead of general natural language.  Overall, the reviewers found the investigation interesting, but too limited for publication at this point.  Given that GPT4 performs well in many settings while handling natural language, it is not clear what is the benefit of restricting the investigation to symbolic data.  Furthermore the paper advertises a general investigation of the axioms of causality, but most of the paper focuses exclusively on transitivity, except for Section 5.  Reviewers would like Section 5 to be expanded with more details and a broader investigation.  Overall, this is very interesting work, but perhaps too preliminary to make interesting observations that could help us understand what axioms of causality LLMs may or may not be able to generalize to in natural language.

**Additional Comments On Reviewer Discussion:**

The reviewers discussed the need to broaden the investigation to natural language beyond abstract symbolic reasoning and to a wider set of axioms along the lines of Section 5.

---

### Decision · Program_Chairs · 2025-01-22

Reject